# The BRCT Domain from the Homologue of the Oncogene PES1 in *Leishmania major* (LmjPES) Promotes Malignancy and Drug Resistance in Mammalian Cells

**DOI:** 10.3390/ijms232113203

**Published:** 2022-10-30

**Authors:** Esther Larrea, Celia Fernández-Rubio, José Peña-Guerrero, Elizabeth Guruceaga, Paul A. Nguewa

**Affiliations:** 1ISTUN Institute of Tropical Health, IdiSNA (Navarra Institute for Health Research), University of Navarra, 31009 Pamplona, Navarra, Spain; 2ISTUN Institute of Tropical Health, Department of Microbiology and Parasitology, IdiSNA (Navarra Institute for Health Research), University of Navarra, 31009 Pamplona, Navarra, Spain; 3Bioinformatics Platform, Center for Applied Medical Research, IdiSNA (Navarra Institute for Health Research), University of Navarra, 31009 Pamplona, Navarra, Spain

**Keywords:** leishmaniasis, cancer, BRCT domain, LmjPES1, drug resistance, gene expression, cell survival, oncogene and cellular proliferation

## Abstract

Around 15% of cancer cases are attributable to infectious agents. Epidemiological studies suggest that an association between leishmaniasis and cancer does exist. Recently, the homologue of PES1 in *Leishmania major* (LmjPES) was described to be involved in parasite infectivity. Mammalian PES1 protein has been implicated in cellular processes like cell cycle regulation. Its BRCT domain has been identified as a key factor in DNA damage-responsive checkpoints. This work aimed to elucidate the hypothetical oncogenic implication of BRCT domain from LmjPES in host cells. We generated a lentivirus carrying this BRCT domain sequence (lentiBRCT) and a lentivirus expressing the luciferase protein (lentiLuc), as control. Then, HEK293T and NIH/3T3 mammalian cells were infected with these lentiviruses. We observed that the expression of BRCT domain from *LmjPES* conferred to mammal cells in vitro a greater replication rate and higher survival. In in vivo experiments, we observed faster tumor growth in mice inoculated with lentiBRCT respect to lentiLuc HEK293T infected cells. Moreover, the lentiBRCT infected cells were less sensitive to the genotoxic drugs. Accordingly, gene expression profiling analysis revealed that BRCT domain from LmjPES protein altered the expression of proliferation- (*DTX3L*, *CPA4*, *BHLHE41*, *BMP2*, *DHRS2, S100A1* and *PARP9*), survival- (*BMP2* and *CARD9*) and chemoresistance-related genes (*DPYD, Dok3*, *DTX3L*, *PARP9* and *DHRS2*). Altogether, our results reinforced the idea that in eukaryotes, horizontal gene transfer might be also achieved by parasitism like *Leishmania* infection driving therefore to some crucial biological changes such as proliferation and drug resistance.

## 1. Introduction

According to the International Agency on Research for Cancer (IARC), 13–15% of cancer cases are attributable to infectious agents, from which two thirds occurred in less developed countries [1,2]. This causal association has been detected with viruses such as the Epstein–Barr virus, hepatitis B and C viruses, human papilloma virus, human T-cell lymphotropic virus type 1, and also with bacteria like *Helicobacter pylori* and parasites such as *Schistosoma haematobium* [2,3,4]. Infectious agents are thought to cause pathological alterations including DNA mutations, cell cycle modulation, dysregulation of DNA repair mechanisms, chronic inflammation and immune system impairment, favoring tumorigenesis [4]. In this context, *Leishmania* parasites have also been associated with several of these cellular pathological changes [5,6].

Leishmaniasis is a neglected tropical disease caused by an intracellular parasite, belonging to the genus *Leishmania,* transmitted to vertebrates including humans by the bite of sandflies, mainly *Phlebotomus* and *Lutzomyia* species. The prevalence of human leishmaniasis has been estimated between 12–15 million cases worldwide, and 1–1.5 million new cases occur each year [7]. According to the clinical manifestations, it can be divided into cutaneous, muco-cutaneous, and visceral or kala-azar [7,8].

Some causal relationships between leishmaniasis and malignancy have been evidenced in experimental animals and in humans [9]. In fact, four different association between these diseases have been described: (a) leishmaniasis mimicking malignancy [10,11,12,13]; (b) leishmaniasis arising from difficulty diagnosing and treating infection among patients receiving chemotherapy for various malignant disorders [14,15]; (c) simultaneous diagnosis of leishmaniasis and a neoplastic disorder in the same tissue samples of patients [16,17,18]; and (d) direct involvement of *Leishmania* spp. in the pathogenesis of malignant lesions, especially skin and mucous membrane tumors [19,20,21,22]. In addition to these associations, the similarities between tumor cells and parasites were largely mentioned when studying expressed proteins and potential therapeutic targets [6]. Furthermore, the availability of *Leishmania* genome allowed the recent identification of the homologue of the oncogene *PES1* named *LmjPES* in *L. major* [23]. In these parasites, *LmjPES* gene was involved in pathogenesis. *LmjPES* overexpressing parasites showed increased in vitro infection ratios as well as higher and faster footpad inflammation in BALB/c mice compared to non-overexpressing parasites [23]. In other organisms, PES1 is known to be implicated in physiological processes like ribosomal biogenesis [24], embryonic development [25], cell proliferation [26] and gene transcription [27]. In addition, it has also been involved in pathological processes, playing a key role in carcinogenesis and tumor progression as shown in different cancer types. In fact, the inactivation of PES1 and its partners involved in ribosome biogenesis demonstrated to be linked to chromosomal instability observed in malignant tumors [28]. On the other hand, it is well known that inherited mutations affecting ribosome biogenesis are related to elevated cancer risk [29]. In breast cancer cells, *PES1* expression has been proved to be elevated; furthermore, proliferation and tumorigenicity were inhibited when *PES1* levels were downregulated [30]. PES1 protein was also dramatically elevated in malignant human astrocytoma, although it was not expressed in differentiated astrocytes [31]. Moreover, in colorectal cancer, other authors not only reported *PES1* expression to be elevated but also demonstrated that *PES1* silenced cells were more sensitive to drug-induced growth inhibition [32]. This nuclear protein contains a carboxyl-terminal domain of the breast cancer gene 1 (BRCT) that was previously found in several proteins involved in cell-cycle checkpoints and DNA repair [31,33,34,35]. It has been reported that the integrity of the BRCT domain in some proteins, was truly important in cell homeostasis and proliferation. In fact, mutations within the BRCT domains of the breast cancer 1 protein (BRCA1) caused breast and ovarian cancer development [36]. In addition, mutations within the BRCT domain tandem of nibrin (NBN) were associated with an increase susceptibility to cancer development [37]. The BRCT domain of PES1 is essential for its role in ribosome biogenesis [24,38]. Holzel *et al*., described point-mutations in the BRCT domain of PES1 causing nucleoplasmic distribution of the protein, and which were similar to cancer predisposing mutations in BRCA1 [38]. Thus, several data confirmed that BRCT domain from PES1 protein exhibited a functional importance in cancer development. In *Leishmania*, LmjPES harbors the firstly identified BRCT domain in these parasites, which has demonstrated to be a promising therapeutic target [39].

Alternatively, the study of horizontal gene transfer (HGT) is emerging. This biological process, consisting in the exchange of genetic material between or among lineages, may take place among prokaryotes as well as eukaryotes. In eukaryotes, HGT can be achieved by viral infection, symbiosis/phagocytosis and parasitism. It has been shown that viruses can carry foreign genes and transfer them between parasites and their hosts, especially retroviruses, which have been identified within certain protozoan parasites such as *Leishmania*, *Trypanosoma, Entamoeba* or *Giardia* [40]. The role of such retroviruses in the regulation of the severity of muco-cutaneous leishmaniasis has been widely studied [41,42,43]. However, the involvement of such viruses as intergenome carriers needs to be studied more deeply. As an illustration in trypanosomes, the integration of minicircles DNA from *Trypanosoma cruzi* into the retrotransposable LINE-1 of various chromosomes that were transferred vertically to their progeny has been also detected [44,45]. Further studies would elucidate the impacts of the integration of genetic material from parasites into the human genome and, consequently, the alterations of clinical manifestations or treatment response. Considering the role of BRCT domains in cancer development, and its presence in *Leishmania* genome, the current studies allow to predict the possible implications of the transfer of BRCT domain from *Leishmania* parasites to their hosts including humans.

## 2. Results

### 2.1. Generation of a Lentivirus Harboring the BRCT Domain Sequence from LmjPES Gene

To study the possible implication of the BRCT domain from LmjPES protein in mammal cells, we generated lentiviral particles expressing either this BRCT domain (lentiBRCT) or the luciferase protein (lentiLuc) and ZsGreen protein. First, we constructed a lentiviral transfer plasmid that contained the coding sequences of both fragments, the BRCT domain from LmjPES and the ZsGreen protein, separated by an IRES sequence (pHIV-BRCT-ZsGreen). Then, using the corresponding transfer plasmid (pHIV-BRCT-ZsGreen or pHIV-Luc-ZsGreen), we generated lentiviral particles to infect both, HEK293T and NIH/3T3 mammal cells. After one-week post-infection, we observed many ZsGreen (+) cells under the fluorescence microscope (Figure 1A). To obtain cultures exhibiting similar cell infection ratios with both lentiviruses, we isolated the ZsGreen (+) cells using a cell sorter. For each cell line (HEK293T and NIH/3T3), we then collected two different pools of cells previously infected with lentiBRCT (BRCT-A and BRCT-B) and one pool of lentiLuc-infected cells. The infection efficiency was greater than 90% of ZsGreen (+) cells through the experiments (Figure 1B). To ensure that those cell pools, in addition to expressing ZsGreen protein, also expressed the BRCT domain from LmjPES protein, we studied in our clones the mRNA expression levels of this BRCT domain. Figure 1C illustrated such expression levels in both cell pools from the cell lines NIH/3T3 and HEK293T (Figure 1C). All the experiments carried out in this work were performed with these pools of cells and interestingly more than 90% of the cells expressed the BRCT domain from LmjPES protein or expressed the Luc protein (controls).

### 2.2. The Expression of BRCT Domain from LmjPES Dramatically Increased the Replication of Mammal Cells

To study the replication of HEK293T and NIH/3T3 cells after the infection with either lentiBRCT or lentiLuc, we analyzed the ATP levels produced by those cells. For this purpose, we seeded 10^4^ cells/100 µL medium in 96-well plates, and two days later ATP levels were quantified. In both HEK293T and NIH/3T3 cells (Figure 2), there was a significantly higher amount of ATP in lentiBRCT-infected cells compared to controls (lentiLuc-infected cells).

Additionally, the proliferative capacity of these cells was determined by flow cytometry assay using the Click-iT EdU kit. We observed a slight accumulation of cells in phases S and G2/M, whereas the percentage of cells in phase G0/G1 was lower in lentiBRCT- vs. lentiLuc-infected cells (Figure 3A,B). These data were consistent with the increased replicative capacity of mammalian cells expressing the BRCT domain from LmjPES protein. Interestingly, when cells were cultured in a limited nutrient medium, we found highest percentages of lentiBRCT-infected cells (Figure 3C,D) in G0/G1 and a lowest percentage in G2/M. These results demonstrated that the expression of BRCT domain from LmjPES significantly altered the cell cycle progression in lentiBRCT-infected cells nutrient-limited in accordance with the greater amount of energy required in actively dividing cells.

### 2.3. The Expression of BRCT Domain from LmjPES Protein Promoted Colony Formation in Mammalian Cells

We analyzed cell survival by measuring the ability of a single cell to generate a colony. This assay was performed seeding one thousand cells (HEK293T and NIH/3T3) infected with lentiBRCT or lentiluc/2 mL medium, in 6-well plates and one week later, we counted the number of colonies in the plates. We observed a significant increase in the number of colonies from lentiBRCT-infected cells vs. controls (Figure 4).

### 2.4. The Studied BRCT Domain Reinforced Tumorigenesis Process In Vivo

Since BRCT domain from LmjPES protein promoted cell proliferation and survival, we therefore speculated that this BRCT domain might induce tumorigenesis. To validate our hypothesis, tumorigenicity assays were performed in vivo. To achieve this goal, 6 × 10^6^ HEK293T cells previously infected with lentiBRCT or lentiLuc were subcutaneously inoculated into the right back of athymic nude mice and the tumor size was measured weekly (Figure 5). Two consecutive assays were completed and we then analyzed the evolution of tumor growth during several days in two different mouse models (inoculated with lentiBRCT or lentiLuc HEK293T infected cells). As observed in Figure 5, at day 29 post-inoculum, the tumor growth was significantly faster in mice harboring lentiBRCT-infected HEK293T cells. These results pointed out to a potential tumorigenic effect of BRCT domain from LmjPES protein.

### 2.5. These Novel Tumorigenic Cells Were More Resistant to Death Induced by Antitumor Drugs

We analyzed the percentage of cell viability, after treating the new cell lines (NIH/3T3 and HEK293T cells infected with lentiLuc or lentiBRCT) during 48 or 72 h with increasing amounts of several antitumor drugs (etoposide, 5-FU or cisplatin). As shown in Figure 6, increasing concentrations of drugs decreased the percentage of cell viability. Furthermore, the novel tumorigenic cells were more resistant to different antitumor drugs such as 5-FU (Figure 6A,D) and etoposide (Figure 6E). No difference was observed in the percentage of cell survival after cisplatin treatment (Figure 6C,F). The IC_50_ values for each drug were analyzed after treating our new cell lines (Table 1). We found significantly higher IC_50_ values for etoposide (NIH/3T3) and 5-FU (NIH/3T3 and HEK293T) in lenti-BRCT-infected cells compared to lentiLuc-infected cells. These data showed that the expression of BRCT domain from *L. major* homolog of PES protein induced death resistance in mammal cells.

### 2.6. Genes Deregulated through BRCT Domain Expression Mainly Overlapped with Metabolic Disease, Cellular Proliferation, Survival and Drug Metabolism Profile

In order to gain a deeper understanding of the molecular mechanisms involved in the cellular alterations described above, after BRCT domain expression, we performed a high-throughput RNA sequencing of LentiBRCT- and Lentiluc-infected HEK293T cells. The data analysis, based on a B statistic cut off (B > 0) and a change in the |log_2_FC (lentiBRCT/lentiLuc)|>1, showed 192 dysregulated mRNAs in lentiBRCT-infected HEK293T cells compared to lentiluc-infected cells. These differentially expressed genes included 72 up-regulated and 120 down-regulated (Figure 7A). Then, we studied these dysregulated genes with the IPA. Examining the molecular categories enriched in our analysis, we found metabolic disease, cancer, nucleic acid metabolism, cellular growth and proliferation, cellular movement, cell death and survival and drug metabolism (Figure 7B). Each of these categories included different cellular functions. In Table 2, we showed the cellular function within each category containing the largest number of dysregulated genes, as well as such genes (in bold type, genes up-regulated; in normal letter, genes down-regulated). The deregulated expression of these genes supported the results showing the alteration of the cellular processes aforementioned in our novel tumorigenic cells.

To further validate the gene expression patterns detected by RNA-Seq analysis, the mRNA expression of 16 randomly selected dysregulated genes (5 up-regulated and 11 down-regulated) from all cellular functions, were characterized by qRT-PCR (Genes underlined in Table 2) (Figure 8), supporting the RNA-seq results.

Overall, these results showed that the expression of the BRCT domain from LmjPES protein in mammalian cells deregulated mRNA levels of genes (including *DTX3L*, *CPA4*, *BHLHE41*, *BMP2*, *DHRS2, S100A1*, *PARP9*, *BMP2, CARD9, DPYD, Dok3*, *DTX3L*, *PARP9* and *DHRS2*) related to cellular processes (such as proliferation and survival) that promote chemoresistance and tumor development.

## 3. Discussion

Although the association between leishmaniasis and certain types of cancer has been observed [9,16,17,18,19,20,21,22]. the exact mechanisms involved have not been described yet. This study was focused on the role played by an oncogenic protein domain from a parasite in the host cell. PES protein has been well characterized in other organisms and is known to have a cell-cycle regulatory function [46] and to play a key role in pathological processes, such as cancer, in which cell proliferation is affected [28,30,31,32]. In *Leishmania* parasites, the homologue of PES, LmjPES, has been recently related to the course of the infection [23]. PES protein including its homologue in *Leishmania* parasites bears a BRCT domain which has also been described to have a functional importance in some carcinogenesis development [36,47]. Thus, we were prompted to assess in two mammal cell lines, NIH/3T3 and HEK293T, whether the expression of BRCT domain from LmjPES protein could induce changes in some biological processes such as proliferation, tumorigenesis and chemotherapeutic-sensitivity profiles. These cell lines have frequently been used for the analysis of the tumorigenic potential of various patterns of gene expression [48,49].

To achieve this aim, we first constructed a lentivirus carrying the BRCT domain from LmjPES protein and we infected NIH/3T3 and HEK293T mammal cells. This strategy allowed a stable and long-term gene expression in almost all cells [50]. In addition, this approach was mimicking a potential horizontal gene transfer (HGT) from the parasite to the host. In fact, HGT has been described between pathogens and host cells [40]. We observed that BRCT domain from LmjPES protein induced in mammalian cells a high capacity of proliferation and survival as well as an alteration of the cell cycle. Our results also illustrated a disruption in cell cycle progression in nutrient-limited BRCT domain expressing cells according to the high level of energy required in actively dividing cells [51]. Besides, the BRCT domain from LmjPES protein conferred higher cell survival ability to both cell lines. The in vivo assays further confirmed the critical role of BRCT domain in the tumorigenesis process and it particularly allowed a high tumor growth. Altogether, these findings indicate that BRCT expression from LmjPES protein may contribute to tumorigenesis by upregulating the capacity of proliferation and survival of cells harboring this domain. Similarly, to these results, the complete *PES 1* gene from mouse previously demonstrated its ability to transform human fibroblasts, by conferring them the capability for colony formation in soft agar [28].

In addition, the RNA-seq analysis revealed an association between the dysregulated genes and the molecular categories: metabolic disease, cancer, cell cycle, cellular growth and proliferation, cell death and survival, nucleic acid metabolism and drug metabolism. In this regard, our transcriptomic analyses provided valuable mechanistic insights. Thus, genes such as *deltex-3-like* (*DTX3L*), *carboxypeptidase A4* (*CPA4*), *basic helix-loop-helix family member e41* (*BHLHE41*), *bone morphogenic protein 2* (*BMP2*), *dehydrogenase/reductase member 2* (*DHRS2*) and *poly (ADP Ribose) polymerase family member 9* (*PARP9*), that have been described to play an important role in promoting cell proliferation and cell cycle regulation [52,53,54,55,56,57,58,59,60], were found significantly deregulated in BRCT domain expressing cells. We also observed a down-regulation of several mitochondrial genes in BRCT domain expressing cells triggering a mitochondrial dysfunction. Our findings are in line with previous works which describe the involvement of the mitochondria in cell proliferation and carcinogenesis [61,62,63].

On the other hand and based on the implication of BRCT domains in DNA damage repair [64,65], drug sensitivity profiles of our BRCT domain expressing cells were assessed using different genotoxic agents: 5-FU, etoposide and cisplatin. In fact, 5-FU, a nucleoside metabolic inhibitor [66]; as well as the etoposide, a topoisomerase inhibitor [67] and cisplatin, an inhibitor of cell division [68], are cytostatic drugs with well-known genotoxic effects [69]. The expression of BRCT domain from LmjPES protein increased drug resistance in mammalian cells and were in accordance with previous data showing a higher chemotherapeutic drug sensitivity in PES-ablated cells [32]. Interestingly, we also demonstrated a higher expression of *dihydropyrimidine dehydrogenase* (*DPYD*), a catabolic enzyme that breaks a large amount of 5-FU into its inactive metabolite [70]. This *DPYD* overexpression may lead to 5-FU resistance due to the higher DPYD-mediated degradation of 5-FU. Indeed, in vitro studies have revealed that *DPYD* overexpression in cancer cell lines correlated with 5-FU resistance [71] and high levels of *DPYD* mRNA expression in colorectal cancer have also been demonstrated to be linked with resistance to 5-FU [72]. Furthermore, we detected mitochondrial cytopathy as one of the cell functions enriched in dysregulated genes in BRCT domain expressing cells. It is well-known that mitochondrial dysfunction can promote chemoresistance [73]. In addition, the deregulated genes found in our transcriptomic analyses as *tyrosine kinase member 3* (*Dok3*), *DTX3L*, *PARP9* and *DHRS2* have been described to be involved in chemoresistance [27,52,58,60]. Chemotherapeutic drugs such as miltefosine have shown effectiveness in leishmaniasis treatment and overall, these results point to a role of BRCT domain from LmjPES protein in drug sensitivity. Even though future studies are needed.

The current work shed some light on the interaction parasite-host. Firstly, the idea of a parasite’s protein domain with the capability of inducing host cell proliferation and survival, thus predisposing to cancer development [27]. And secondly, the possibility of this same domain transfer from an eukaryote parasite to eukaryote host cells [40], conferring them a chemo-resistance profile. The current study shows the possible implication of HTG between parasites and hosts in the development of certain types of cancer as well as in their successful treatment.

Altogether, this study provides information about *Leishmania*-host interaction, suggesting an implication of LmjPES protein in human cancer development through its BRCT Domain. In fact, our data demonstrated that such a domain promoted malignancy and drug resistance in mammalian cells by deregulating mRNA levels of several genes (such as *DTX3L*, *CPA4*, *BHLHE41*, *BMP2*, *DHRS2, S100A1*, *PARP9*, *BMP2, CARD9, DPYD, Dok3*, *DTX3L*, *PARP9* and *DHRS2*) related to proliferation, survival and other cellular processes. Future studies are needed to better understand the role of the aforementioned genes during leishmaniasis progression. This will also allow to establish a better relationship between leishmaniasis and cancer, and to further discover and exploit new therapeutic targets.

## 4. Materials and Methods

### 4.1. Cell Culture

The human cell line HEK293T (ATCC CRL-1573) and the mouse cell line NIH/3T3 (ATCC CRL-1658) were cultured in Dulbecco’s modified Eagle’s medium (DMEN) supplemented with 10% (*v*/*v*) heat-inactivated fetal bovine serum (FBS), 100 IU/mL penicillin and 100 μg/mL streptomycin (all culture reagents from Gibco Laboratories, Grand Island, NY, USA). Cells grew at 37 °C in a humidified atmosphere containing 5% CO_2_. All cell lines were routinely tested for *Mycoplasma*.

### 4.2. Lentiviral Vector Assembly

The lentiviral transfer plasmid, pHIV-BRCT-ZsGreen, was generated by introducing the BRCT domain sequence from *LmjPES* gene, into pHIV-Luc-ZsGreen vector (Addgene, Watertown, MA, USA). This vector includes the coding sequence for luciferase (Luc) and ZsGreen proteins separated by an Internal ribosome entry site (IRES) element. To replace the Luc sequence from pHIV-Luc-ZsGreen vector by the BRCT domain sequence from *LmjPES* gene, the pHIV-Luc-ZsGreen vector was digested overnight with NotI and BamHI restriction enzymes (New England Biolabs, Ipswich, MA, USA). Next, the pHIV-ZsGreen linearized vector was purified from an agarose gel with NucleoSpin Gel and PCR Clean-up kit (Macherey-Nagel, Düren, Germany) following manufactured instructions. BRCT domain sequence from *LmjPES* gene was amplified by polymerase chain reactions (PCR) from *L. major* total DNA using the primers, sense 5’-GTGTCGTGAGCGGCC*CCACC*ATG**CGCGGGCTAACCTTCTTCATATC**-3’ and antisense 5’-GGAGAGGGGCGGATCTCA**GCGGTAGCCCGTCACCGGG**-3’. These primers were designed with the In-Fusion cloning software (Takara, Shiga, Japan) and they included, the sequences of the ends of the BRCT domain (bold), start and end codons of the translation (underlined), kozak sequence (italics) and the sequence of the ends of NotI and BamHI digested pHIV-ZsGreen vector. The obtained PCR product was also purified with NucleoSpin Gel and PCR Clean-up kit after running an agarose gel electrophoresis. Then, this BRCT domain sequence from *LmjPES* gene was cloned into linearized pHIV-ZsGreen vector using the In-Fusion cloning kit (Clontech, Montain View, CA, USA). This adenoviral transfer vector containing the BRCT domain sequence from *LmjPES* gene was named thereafter pHIV-BRCT-ZsGreen. The accurate insertion of BRCT domain sequence in this vector was verified by PCR and sequencing.

After pHIV-BRCT-ZsGreen adenoviral transfer vector construction, we generated lentiviral particles carrying this BRCT domain sequence. For that, HEK293T cells were seeded at 8 × 10^6^/150 cm^2^ culture dish in DMEM medium supplemented with 10% FBS and antibiotics. Twenty-four hours later, cells were co-transfected with three plasmids: the packaging plasmid that encodes for *gag*, *pol*, rev and *tat* genes (pA3), the envelope plasmid expressing the glycoprotein G of the Vesicular Stomatitis Virus (VSV-G) (pVSV.G) and the transfer vector pHIV-BRCT-ZsGreen or pHIV-luc-ZsGreen. The transfection was performed using a ratio of plasmid mass of 12:6:18, respectively, and polyethylenimine (PEI) (Sigma Aldrich, St. Louis, MO, USA) at a 10:1 ratio of PEI/DNA. Viral supernatants, containing lentiviral particles expressing the BRCT domain from *LmjPES* gene (lentiBRCT) or expressing the Luc protein (lentiLuc) were harvested after 72 h, clarified and filtered through 0.45 μm pore size and kept at −80 °C to further experiments.

### 4.3. Lentiviral Transduction of Mammal Cells

Human HEK293T and mouse NIH/3T3 cells were seeded in 6-well plate at 1.5 × 10^4^ cells/2 mL medium per well. After 24 h, cells were transduced with lentiBRCT. Control cells transduced with lentiLuc was also performed. The transductions were carried out with 1 mL of previously generated viral supernatants in the presence of polybrene (10 μg/mL, Santa Cruz Biotechnology, Dallas, TX, USA). Then, cells were monitored by fluorescent microscopy for ZsGreen production, an indicator of the infected cell number. To achieve nearly 100% infection of the cells, ZsGreen positive cells were purified by flow cytometry (BD FACSII sorter).

### 4.4. Cell Proliferation Assay

To analyze the rate of cell proliferation, Cell Titer-Glo Luminescent Cell Viability Assay Titer-GLO kit (Promega, Madrid, Spain) was performed. Cells were seeded at a density of 10^4^ cells/100 µL medium per well in 96-well plates (sextuplicates) and leaved out for 48 h. ATP levels in the medium were measured as the manufacturer recommended.

### 4.5. Cell Cycle Analysis

The cellular DNA content was determined by flow cytometry assay using Click-iT EdU kit (Invitrogen™/Molecular Probes, Vilnius, Lithuania) following the manufacturer’s instructions. To measure the DNA content, the cells were treated with RNase A (200 μg/mL; Sigma-Aldrich) at 37 °C for 1 h and stained with propidium iodide (75 μg/mL, Sigma-Aldrich). Samples were immediately analyzed using a FACSCanto flow cytometer (BD Biosciences, Heidelberg, Germany) and the FACSDiva software v8.0.1 (BD Biosciences). The percentage of cells in G0/G1, S, and G2/M cell cycle phases was obtained using FlowJo software v10. The study of cell division was performed in cells under normal growth conditions (DMEN plus 10% FCS) and, in nutrient-limited conditions for 24 h (DMEN plus 2% FCS).

### 4.6. Colony Formation Assay

Colony formation assays were performed with HEK293T and NIH/3T3 cells infected with lenti-BRCT or Lenti-Luc particles. One thousand cells were seeded in 2 mL complete medium in 6-well plates. Cultures were maintained one week. Plates were washed with PBS, fixed with methanol (Scharlab, Barcelona, Spain) for 20 min and stained with crystal violet (Sigma-Aldrich). Representative pictures were taken. At least two replicates were performed.

### 4.7. Drugs Resistance Study

To determine whether the expression of BRCT domain from LmjPES protein conferred resistance to death in mammalian cells after drug treatments, we analyzed the cell viability of NIH/3T3 and HEK293T cell lines expressing or not expressing this BRCT domain, after 5-fluorouracil (5-FU), (Sigma-Aldrich), etoposide (Sigma-Aldrich) or cisplatin (Sigma-Aldrich) treatments. For that, 10^4^ cells were seeded in 96-well plate and after 24 h, increasing concentrations of drugs were added (5-FU from 0.02 to 8000 µM, etoposide from 0.06 to 1000 µM and cisplatin from 4 × 10^−4^ to 166 µM. After 48 h (etoposide) or 72 h (5-FU and cisplatin) of incubation, the half-maximal inhibitory concentration (IC_50_) was calculated. The decrease in the number of viable cells was determined using the MTT colorimetric assay as previously described [74].

### 4.8. In Vivo Tumorigenicity Assay

All the procedures with animals were approved by the Animal Care Ethics Commission of the University of Navarra (CEEA) and by the institutional ethics committees (Ref R-103-18GN). To generate xenograft tumors, 6 × 10^6^ lentiBRCT or lentiLuc HEK293T infected cells were subcutaneously injected into the right back of 6-week old athymic nude female mice (Harlan, Barcelona, Spain) in 100 µL of saline solution. The tumor size was measured weekly with a caliper. The assay was performed twice consecutively, the first time, with nine animals (four lentiLuc HEK293T infected cells and five LentiBRCT HEK293T infected cells) and the second time, with 12 animals (six lentiLuc HEK293T infected cells and six LentiBRCT HEK293T infected cells). All the animals were sacrificed on day 29 after the cell injection, when the tumor volume exceeded 3000 mm^3^ in some of the mice. Tumor volumes were calculated using the formula: tumor volume (mm^3^) = 0.52 × (length)^2^ × (width).

### 4.9. RNA Preparation and RNA-Sequencing (RNA-seq) Analysis

Total RNA from lentiBRCT and lentiLuc HEK293T infected cells was extracted using the automated Maxwell system (Promega). First, the quality control of the total RNA samples was performed to verify that it met the required standards for total RNA quantity, RNA integrity number (RIN) value and rRNA ratio. Then, the samples were shipped on dry ice to Macrogen Company (Seoul, Korea) where they were processed and sequenced. The sequencing library from mRNA was prepared using TruSeq Stranded mRNA Library Preparation Kit (Illumina, San Diego, CA, USA) and RNA-seq was carried out using NovaSeq6000 platform following a configuration of paired-end of 100 base pairs. RNA-seq data analysis was performed using the following workflow: The quality of the samples was verified using FastQC software. The alignment of reads to the human genome (hg38) was performed using STAR [75]. Gene expression quantification using read counts of exonic gene regions was carried out with featureCounts [76]. The gene annotation reference was Gencode v31 [77]. Differential expression statistical analysis was performed using R/Bioconductor [78]. Data were publicly available in GEO database with the accession number GSE160853.

First, gene expression data was normalized with edgeR [79] and voom [80]. After quality assessment and outlier detection with R/Bioconductor [78], a filtering process was performed. Genes with read counts lower than six in more than the 50% of the samples of all the studied conditions (lentiBRCT and lentiLUC) were considered as not expressed in the experiment under study. LIMMA (Linear Models for Microarray Data) [80] was used to identify the genes with significant differential expression between experimental conditions. Genes were selected as differentially expressed using a B cut off B > 0 and |log_2_FC|>1. Further functional and clustering analyses and graphical representations were performed using R/Bioconductor [78]. The biological knowledge extraction was complemented through the use of Ingenuity Pathway Analysis (IPA) (https://www.qiagenbioinformatics.com, accessed on 24 September 2019), which database includes manually curated and fully traceable data derived from literature sources.

### 4.10. Quantitative Real-Time Polymerase Chain Reaction (qRT-PCR)

The expression levels of dysregulated genes in lentiBRCT HEK293T infected cells were further validated by qRT-PCR. Total RNA from cell lines was extracted using the automated Maxwell system. Reverse transcription was performed as previously reported [81]. Real-time PCR were performed with iQ SYBR Green supermix (Bio-Rad, Hercules, CA, USA) in a CFX96 Real-Time system (Bio-Rad), using specific primers for each gene (Table 3). The amount of transcript was quantified by the formula 2^ct(house-keeping gene)-ct(gene in study)^, with ct being the point at which the fluorescence rises appreciably above the background fluorescence [81]. The house-keeping gene used in this study was β-actin.

### 4.11. Statistical Analysis

Statistical significance was assessed using Prism v5 (GraphPad) or Stata v12. We used Mann-Whitney’s U test for non-normally distributed variables. Data were presented as mean ± SD. The in vivo tumor growth was analyzed with a multilevel mixed-effects linear regression model followed by linear contrast. These data were presented as the mean ±95% confidence interval (CI). Statistical significance was set at *p* < 0.05 (*) and *p* < 0.01 (**).

RNA-seq statistical analysis was performed using R/Bioconductor and LIMMA as described above.

## Figures and Tables

**Figure 1 ijms-23-13203-f001:**
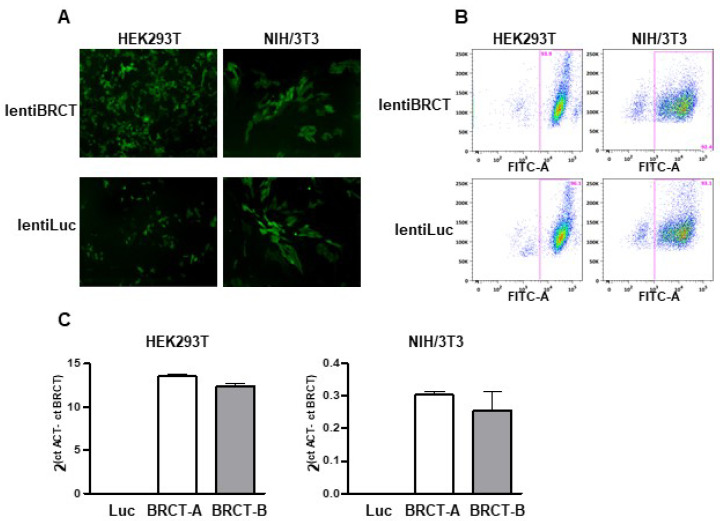
Production and verification of a lentivirus carrying the BRCT domain from *LmjPES*. To study the implication of BRCT domain from LmjPES protein in mammal cells, we first performed lentiviral particles expressing this BRCT domain (lentiBRCT) or the Luciferase protein (lentiLuc) and the ZsGreen protein. (**A**) Image obtained with the fluorescence microscope of HEK293T and NIH/3T3 cells infected with both lentiviruses, lentiBRCT or lentiLuc. (**B**) Cell sorting analysis of positive ZsGreen lentiBRCT- or lentiLuc-infected HEK293T and NIH/3T3 cells. (**C**) mRNA expression levels of BRCT domain from *LmjPES* molecule in lentiBRCT (BRCT-A, BRCT-B) or lentiLuc (Luc), HEK293T and NIH/3T3 infected cells. The values shown in the (**C**) represent the mean ± SD.

**Figure 2 ijms-23-13203-f002:**
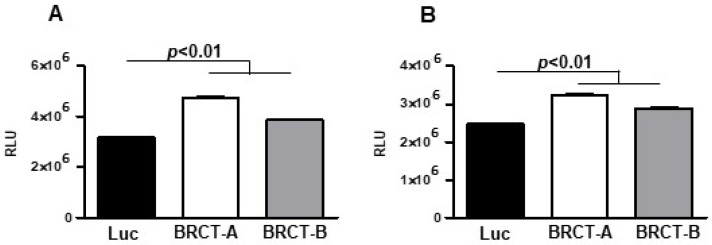
Analysis of BRCT domain from LmjPES protein expression on the capacity of mammal cells to produce ATP. ATP levels (measured as relative light unit: RLU) produced by lentiBRCT (BRCT-A, BRCT-B) or lentiLuc (Luc) infected HEK293T (**A**) and NIH/3T3 (**B**) cells. The values represent the mean ±SD of sextuplicates from a representative experiment of the three carried out.

**Figure 3 ijms-23-13203-f003:**
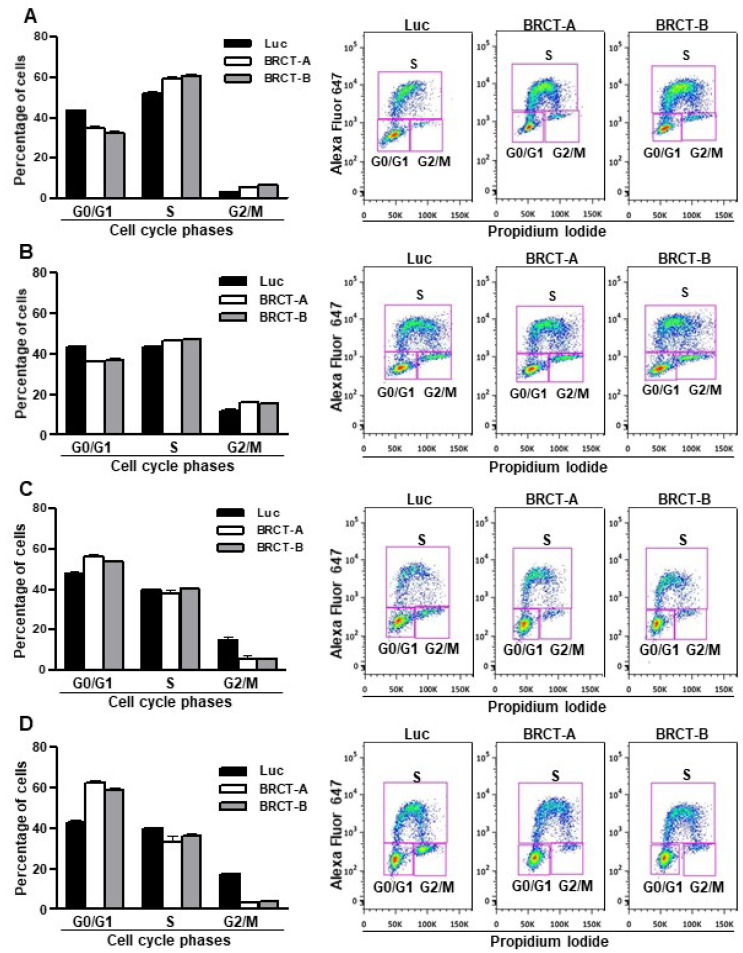
Effect of the expression of BRCT domain from LmjPES protein on cell cycle progression. Percentage of cells in the different phases of cell cycle and representative image of cell cycle analysis from HEK293T (**A**) and NIH/3T3 (**B**) infected cells with lentiBRCT (BRCT-A and BRCT-B) or lentiLuc (Luc). Percentage of cells in the different phases of cell cycle and representative image of cell cycle analysis from HEK293T (**C**) and NIH/3T3 (**D**) cells cultured in a limited nutrient medium and infected with lentiBRCT (BRCT-A and BRCT-B) or lentiLuc (Luc).

**Figure 4 ijms-23-13203-f004:**
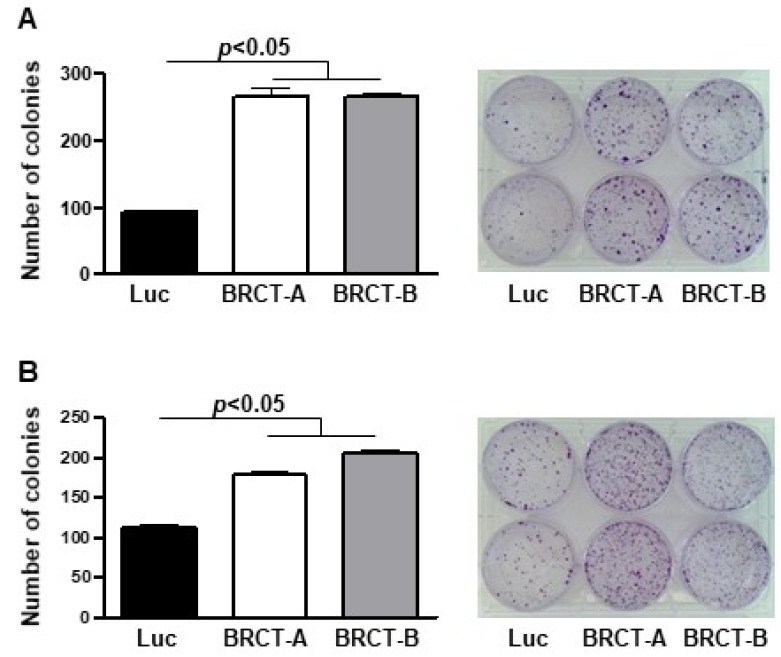
Study of colony formation from mammalian cells expressing the BRCT domain from LmjPES protein. Number of colonies formed after one week of culture of lentiBRCT (BRCT-A and BRCT-B) or lentiLuc (Luc) HEK293T (**A**) and NIH/3T3 (**B**) infected cells. The values represent the mean ±SD of duplicates from three experiments carried out.

**Figure 5 ijms-23-13203-f005:**
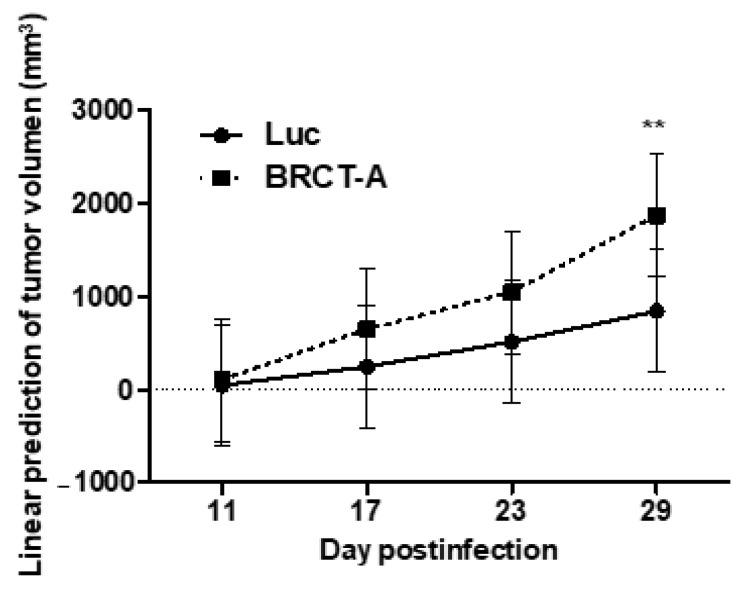
Assay of tumorigenesis in vivo from mammal cells expressing the BRCT domain from LmjPES protein. The graph shows the tumor growth over time of HEK293T cells infected with lentiBRCT (BRCT-A) or Lentiluc (Luc). ** *p* < 0.01, LentiBRCT vs. Lentiluc infected HEK293T cells, at day 29 post-inoculum. The values shown in the figure represent the mean ± SD from one experiment of the two carried out.

**Figure 6 ijms-23-13203-f006:**
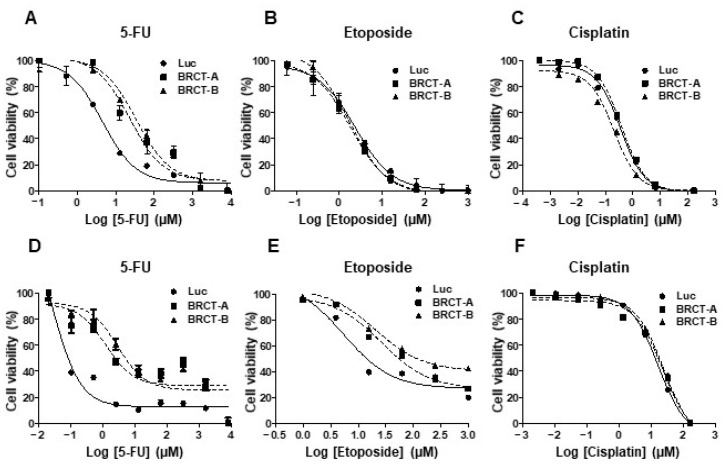
Effect of BRCT domain from LmjPES protein expression on drug-induced cell death. Percentage of cell viability of HEK293T (**A**–**C**) and NIH/3T3 (**D**–**F**) cells infected with lentiLuc (Luc) or lentiBRCT (BRCT-A and BRCT-B) after 48 or 72 h of treatments with increasing amounts of 5-Fluorouracil (5-FU) (**A**,**D**), etoposide (**B**,**E**) or cisplatin (**C**,**F**) drugs. The values represent the mean ± SD of triplicates from three independent experiments carried out.

**Figure 7 ijms-23-13203-f007:**
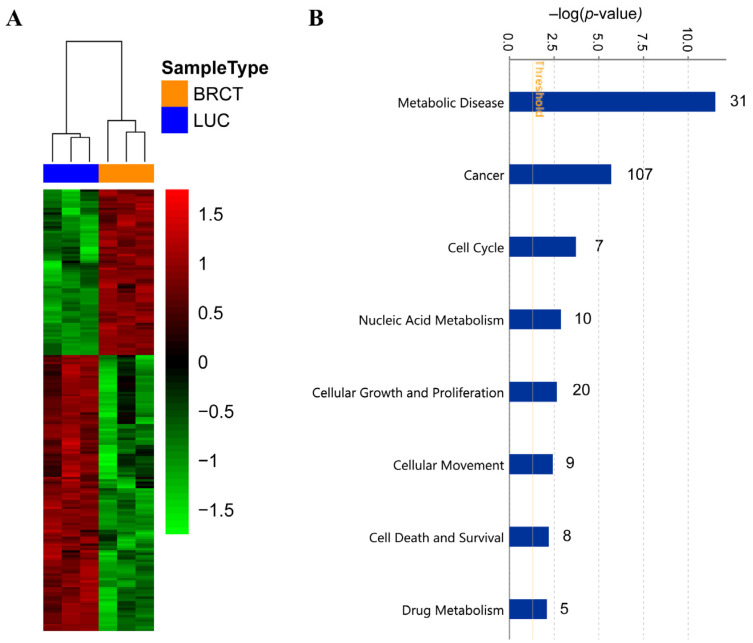
RNA sequencing analysis of LentiBRCT and Lentiluc HEK293T infected cells. Heatmap of the 192 dysregulated genes in lentiBRCT (BRCT) infected HEK293T cells compared to lentiluc (LUC) infected cells. The experiment was performed in triplicate (**A**). Molecular categories from Ingenuity pathway Analysis of the 192 dysregulated genes in lentiBRCT infected HEK293T cells compared to lentiluc infected cells. The graph presents the molecular categories ordered by –log (*p* value) and the number of genes from our list of 192 dysregulated that are in each specific category (in the right part of the bars) (**B**).

**Figure 8 ijms-23-13203-f008:**
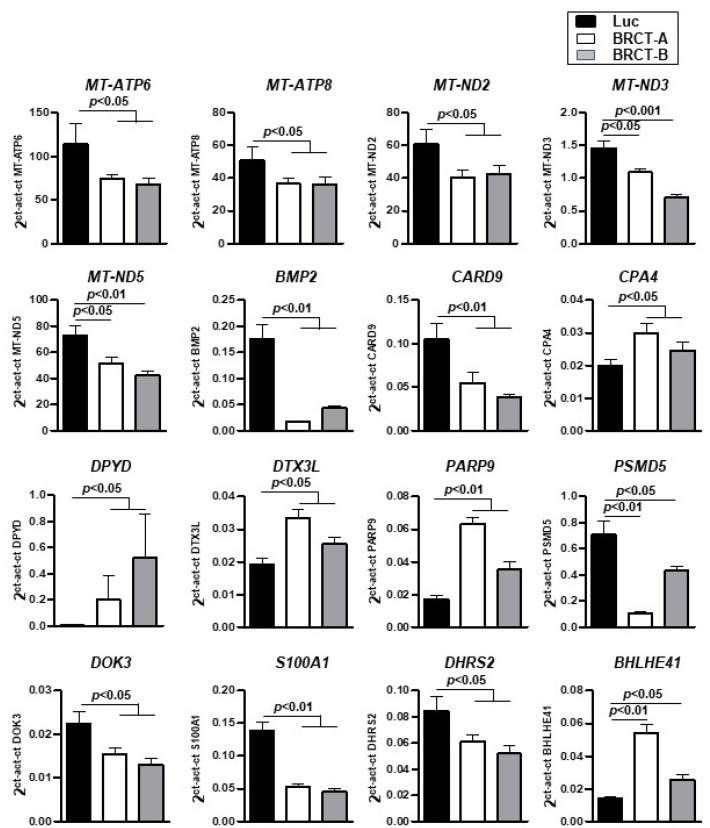
Validation of gene expression by real time-PCR. mRNA expression levels of 16 dysregulated genes found in the RNA-seq analysis and validated by real time-PCR, in HEK293T cells infected with lentiLuc (Luc) or lentiBRCT (BRCT-A and BRCT-B). The values shown in the figure represent the mean ±SD of sextuplicates.

**Table 1 ijms-23-13203-t001:** IC_50_ values of DNA-damaging agents on mammalian cells expressing the BRCT domain from LmjPES molecule (BRCT-A and BRCT-B) or expressing luciferase (Luc) as control.

Condition	5-Fluorouracil (µM)	Etoposide (µM)	Cisplatin (µM)
HEK293T-Luc HEK293T-BRCT-A HEK293T-BRCT-B	3.75 ± 1.06 21.75 ± 2.47 * 29.65 ± 6.58 *	2.43 ± 0.09 1.92 ± 0.11 1.82 ± 0.11	0.58 ± 0.32 0.54 ± 0.23 0.30 ± 0.13
NIH/3T3-Luc NIH/3T3-BRCT-A NIH/3T3-BRCT-B	0.028 ± 0.002 1.26 ± 0.05 * 3.00 ± 0.14 *	6.37 ± 1.29 24.44 ± 2.94 * 19.94 ± 0.78 *	16.82 ± 0.25 21.52 ± 2.15 23.26 ± 1.77

Mean ± SD, * *p* < 0.05 vs. Luc.

**Table 2 ijms-23-13203-t002:** Ingenuity pathway analysis of differentially expressed mRNAs in BRCT domain from LmjPES protein expressing HEK293T cells compared to control cells.

Categories	Diseases or Functions Annotation	Molecules
Metabolic Disease	Mitochondrial cytopathy	* MT-ATP6 * *, MT-ATP8, MT-ND1, MT-ND2, MT-ND3, MT-ND4, MT-ND4L, MT-ND5*
Cancer	Skin cancer	** *ABI3BP* ** *, **ACE**, **ANG**, **B3GALT1**, BMP2, **C1R**, CARD9, **CDH7**, **CHST8**, **CLSTN2**, CNKSR1, **COL14A1**, **COL21A1**, **CPA4**, CPNE7, **CSMD3**, CUL9, **CYP1A1**, **DCHS2**, **DOK6**, **DPYD**, **DTX3L**, **EOMES**, FGF13, FHAD1, FLRT2, FTCD, GABRE, **GALNT5**, **GLIS1**, GNB3, **IFIH1**, **KCNJ4**, **LCP1**, **LDHD**, **LRAT**, **LRRN2**, **MAGEA3/MAGEA6**, **MDM1**, MST1R, MT-ND5, MUC6, NAP1L2, **NELL1**, **NLRP1**, **NPNT**, NPR2, OR51B5, **PARP9**, **PCK1**, **PDE1A**, **PLAT**, PLIN4, **PLXNA4**, PRKCB, **RPGRIP1**, **SLC7A3**, **SLCO2A1**, SNX19, **SPOCK3**, **SYNPO**, THPO, **TOX**, TREX2, **TRPC4**, WDR66, **ZC3H6**, ZNF114, **ZNF331**, **ZNF525***
Cell cycle	Cell cycle progression	* BMP2 * *, THPO*
Nucleic acid metabolism	Metabolism of nucleic acid component or derivative	** *CYP1A1* ** *, **DPYD**, **HAAO**, HAND1, MT-ATP6, MT-ATP8, NPR2, **PDE1A**, PRKCB, PSMD5*
Cellular growth and proliferation	Expansion of cells	*BMP2, DHRS2, DOK3, **EOMES**, **LCP1**, **PLAT**, S100A1, THPO, **TOX***
Cellular movement	Invasion of cancer cell lines	** * BHLHE41 * ** *, MST1R, **PLAT***
Cell death and survival	Cell viability of cancer cell lines	** *CD24* ** *, MST1R, BMP2, THPO*
Drug metabolism	Clearance of 5-fluorouracil	** * DPYD * **

(**In bold type**, genes up-regulated; in normal letter, genes down-regulated; underlined, genes validated by real time-PCR).

**Table 3 ijms-23-13203-t003:** Primers used in this study.

Target Gene	Sense Primer (5’-3’)	Antisense Primer (5’-3’)
*MT-ATP8*	ATGGCCCACCATAATTACCC	GTTCATTTTGGTTCTCAGGG
*PARP9*	GTTAGTTTGCAAGGGAAGCC	AATTACATCTGCCGTCTGCC
*MT-ATP6*	TTTCCCCCTCTATTGATCCC	AGCCTATAATCACTGTGCCC
*MT-ND5*	CATCAGTTGATGATACGCCC	GAAGGGCTATTTGTTGTGGG
*CARD9*	CTGTACAAGAAGGTCACAGG	AGTCATCTTTGGAGCTCAGC
*PSMD5*	GCTTAACGAGAACCATAGGG	CAATCTGGGAAAGAGTGAGG
*S100A1*	CAAGAAGGAGCTGAAAGAGC	TATGGAGAGGGATAAGTGGG
*DTX3L*	GGTGGATACTGTTCATTGGG	TGTCACTGCGTACTCTAAGC
*BHLHE41*	GCAAGAGAGACAGTTACTGG	CCAAGACTACAGCTTTCTCC
*DPYD*	CAACTCTGTGTTCCACTTCG	AGGCATCTCATTGCTTCTCG
*CPA4*	TACTGAAGTTCAGCACTGGG	CATATCCATCAGGATTGGCC
*MT-ND3*	TAATCAACACCCTCCTAGCC	ACTCATAGGCCAGACTTAGG
*MT-ND2*	ATTTCCTCACGCAAGCAACC	TCATGTGAGAAGAAGCAGGC
*DOK3*	TCCTCAGGATCCACAGATGC	CCTCAGTCCTCTGCACCACC
*BMP2*	ACATGCTAGACCTGTATCGC	GTTTTCCCACTCGTTTCTGG
*DHRS2*	ATGCTGTCAGCAGTTGCCCG	CCGATCCCACTGGTGGACCC
*β-actin*	AGCCTCGCCTTTGCCGA	CTGGTGCCTGGGGCG
*LmjBRCT*	TCTTCATATCGCGTGAGGTG	CATGCTTTTTCATCCCTGGC

## Data Availability

The data from RNA-seq were publicly available in GEO database with the accession number GSE160853.

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
