# Peer review of "The BRCT Domain from the Homologue of the Oncogene PES1 in Leishmania major (LmjPES) Promotes Malignancy and Drug Resistance in Mammalian Cells"

_ijms, 2022, doi:10.3390/ijms232113203_

Round 1

Reviewer 1 Report

Find comments in the attachment

Author Response

Thank you.

Reviewer 2 Report

Title-BRCT Domain From The Homologue Of The Oncogene PES1 In Leishmania major (LmjPES) Promotes Malignancy And Drug Resistance In Mammalian Cells by Esther Larrea et al. 2022.

Summary of Article- In the present work, author focuses on the homologue of PES1 in Leishmania major (named LmjPES) and BRCT domain research (DNA damage-responsive checkpoints). Here, authors elucidate the hypothetical oncogenic implication of BRCT domain from LmjPES in host cells via molecular experiments. They found the expression of BRCT domain from LmjPES conferred to mammal cells in vitro a greater replication rate and higher survival. They also found a faster tumor growth in mice inoculated with lentiBRCT HEK293T infected cells respect to lentiLuc HEK293T infected cells. They also showed that that the lentiBRCT infected NIH/3T3 cells were less sensitive to the genotoxic drugs 5-FU and etoposide, and lentiBRCT infected HEK293T cells were less sensitive to the drug 5-FU. Here, BRCT domain from LmjPES protein altered the expression of proliferation-, survival- and chemoresistance-related genes. They suggested that in eukaryotes, horizontal gene transfer might be also achieved by parasitism like Leishmania infection driving therefore to some crucial biological changes such as proliferation and drug resistance.

 Comments- This is a very important systematic study on the Leishmania infection mechanism responsible for proliferation and drug resistance. Article accepted with few small corrections mentioned below-

1.     Abstract content does not match very much to the rest of article. It should be written with other words having the same meaning. Size also checks as per journal.

2.     In Keywords – molecular mechanism related terminology such as gene expression or gene or genetics or related terms incorporated as this is main work defined in the article.

3.     Although the relations of cancer and leishmania are not very sure till today but several common and related mechanisms drive the disease condition. Therefore, if the author describes the future of study based on molecular pathway, then it will conclude the study with further prospect.

Acceptance- Yes

Decision- Minor revision

Author Response

Thank you
